# Impact of Land-Use Changes on Climate Change Mitigation Goals: The Case of Lithuania

Renata Dagiliūtė 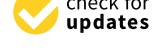 and Vaiva Kazanavičiūtė *

Department of Environmental Sciences, Vytautas Magnus University, Universiteto Str. 10, Akademija,
LT-53361 Kaunas, Lithuania; renata.dagiliute@vdu.lt
* Correspondence: vkazanaviciute@gmail.com

**Abstract:** The land-use, land-use change and forestry (LULUCF) sector is receiving increasing attention in climate change mitigation and greenhouse gas (GHG) emission offsetting. The sector itself and measures applied to mobilize this sector in order to tackle climate change are dominant in nationally determined contributions under the Paris Agreement as well as in national strategies, as in the case of Lithuania. Lithuania has set the goal of becoming a carbon-neutral country in 2050, reducing GHGs by 80% compared to 1990 and offsetting the remaining 20% through the LULUCF sector. Therefore, this paper aims at analyzing historical land-use changes in 1990–2021, as reported for the United Nations Framework Convention on Climate Change (UNFCCC) secretariat, and LULUCF's potential to achieve climate change mitigation goals, taking into account different land-use change scenarios (business as usual, forest development, forest development + additional measures and forest land 40% + additional measures) for 2030 and 2050 in Lithuania. The scenarios are based on historical and potential future policy-based land-use changes. Projections of GHG emissions/removals for different scenarios are prepared according to the Good Practice Guidance and Uncertainty Management in National Greenhouse Gas Inventories (2006) by the Intergovernmental Panel on Climate Change (IPCC). The results indicate that land-use changes over the period 1990–2021 remained rather stable, with some increases in forest area and grassland at the expense of cropland. The whole LULUCF sector acted as a carbon sink in most cases, forests being a key category for removal. However, reaching climate neutrality in 2050 might be challenging, as the goal to offset 20% of remaining GHG emission compared to 1990 through LULUCF would not be met in any of the scenarios analyzed, even the scenario of maximal forest-area development and additional measures. Considering the high historical GHG-removal fluctuations and the uncertainties of the sector itself, caution should be taken when relying on LULUCF's potential to reach the set goals.

**Keywords:** LULUCF; climate change; mitigation; policy; forestry; GHG

## 1. Introduction

In 2021, pursuing the global sustainability agenda [1], particularly goal 13 on climate action, and in line with the Paris Agreement [2] as well as the European Union aims on climate neutrality [3], Lithuania adopted the National Climate Change Management Agenda [4], setting ambitious goals for GHG reduction in the short term and the long term. Compared to 1990, the national agenda foresees reduction of national net GHG emissions by 70% by 2030 and reduction of net GHG emissions by 100% by 2050 [4]. Both goals strongly rely on the LULUCF sector and its carbon removals: the 2030 reductions include absorption in the LULUCF sector (no specific target in the national agenda, but according to EU Regulation 2023/839 [5], removal should amount at least to 4.633 million t $CO_2$ eq for Lithuania); the 2050 goals anticipate an 80% reduction due to various climate change mitigation measures applied in different economic sectors, while the remaining 20% will be offset by the LULUCF sector [4]. It means that 9.558 million t $CO_2$ eq must be absorbed by the LULUCF sector in 2050 in Lithuania.

Lithuania is not the only country to rely significantly on the LULUCF sector while aiming at carbon neutrality. Under the Paris Agreement, countries were required to undertake national commitments for greenhouse gas (GHG) emissions reduction, known as nationally determined contributions (NDC), and prepare long-term low greenhouse gas emission development strategies to achieve climate change policy goals. A recent UN report [6] on NDC indicates that most of the countries included the LULUCF sector and corresponding mitigation measures in their NDC. Most often, this covers afforestation, reforestation and revegetation (48%). Such reliance on the LULUCF sector is supported by a number of studies showcasing high historical carbon removals [7–9] and/or potential net carbon removal [10–12] particularly by forests. For example, Finnish forests could outweigh carbon emissions from the other sectors no later than 2040, as indicated by Kallio et al. [13]. It is also estimated that forest expansion alone could contribute from 6% to 10% of the EU GHG emission reduction target [14]. In addition, the forestry sector plays a role in GHG emission reduction via sequestration not only in biomass but also in forest products due to both storage and substitution effects [15–20]. For example, it is suggested [13] that use of wood for bioenergy would significantly reduce emissions and play an important role in reaching Finland's 80% emission-reduction target by 2050. Nevertheless, the results of a study performed for European forests [21] indicate that to achieve climate neutrality, the EU forests' net carbon removals should increase about 25%, while the combined EU + UK forest sink is projected to decline; therefore, additional efforts are needed.

Hence, next to the forest related measures, other measures might also be significant for carbon removal in the LULUCF sector. Usually, wetland restoration, soil carbon sequestration, bioenergy with carbon capture and storage (BECCS) and agroforestry are analyzed as important land management measures for GHG removals [22,23]. Still, forest-related measures cover the largest share of mitigation potential, followed by peatland restoration and soil organic carbon enhancement on agricultural lands (cropland and grassland) [23]. In general, it is estimated that land-use-based measures have the potential to contribute approximately 20–30% to the 1.5° temperature target before 2050 [24–27]. Though policies and measures in land-use-related sectors in 2009–2019 have contributed only to 0.5% of total emission reduction [23], reliance on the LULUCF sector might even increase, especially if GHG reductions in other sectors fail to be achieved [23,26]. It may also be the case that a focus on LULUCF will decrease efforts to reduce emissions in other sectors [28].

Lithuania has not provided a separate NDC but is represented in the EU's joint NDC [29]. As mentioned, national climate neutrality goals are set in the National Climate Change Management Agenda [24], setting GHG reduction targets for 2030 and 2050, which include LULUCF GHG removals. To ensure the implementation of these commitments, measures for different sectors, including LULUCF, are listed in Policies and Measures and Projections of Greenhouse Gas emissions in Lithuania [30]. Measures for the LULUCF sector include biomass sink enhancement, such as afforestation and reforestation, restoration of damaged forests and redevelopment of shrubs as well as various soil carbon stock enhancement measures: wetland restoration, grassland management in locations with organic soils, promotion of perennial crops, promotion of cultivation of cover crops, and promotion of no-tillage agricultural practices. To date, the main factors and drivers of land-use-related GHG emissions and removals in Lithuania are considered to be land-use changes due to political and economic factors. Land-use changes were induced by the restructuring of the agricultural sector after the restoration of independence, followed by support allocated for rural development after joining the EU, intensive afforestation of abandoned land or land not suitable for agriculture [31,32], strict governmental control of deforestation and preservation of domestic forest resources [33,34].

Therefore, the main research questions of this paper are (i) whether Lithuania can rely on LULUCF and (ii) whether foreseen measures are sufficient to increase GHG removals in the LULUCF sector to reach desirable levels. For that purpose, historical land-use changes and net removals/emissions by the LULUCF sector in Lithuania in 1990–2021 are analyzed,

and based on land-use scenarios and planned measures, the potential of the sector to contribute to the climate change mitigation goals for 2030 and 2050 is estimated.

The paper is structured as follows. The Section 2 introduces the time frames and data sources as well as descriptions of selected land-use scenarios and measures included in the estimations of LULUCF's GHG removal potential. Section 3 presents the main findings on land-use changes, the sector's net GHG emissions and its potential to contribute to the national climate change mitigation goals in the long run. The paper closes with a discussion and conclusions.

## 2. Methods

### 2.1. Land-Use Changes and National GHG Emissions

Historical analysis of land-use changes covers 1990–2021 period and is based on the data collected by the State Forest Service in executing the National Forest inventory (NFI), which serves as the main database for national greenhouse gas inventory and provides data on annual area and its changes covering all land uses—forest land, cropland, grassland, wetlands, settlements and other land. A matrix of land-use changes is developed from the monitoring of more than 16,000 sampling plots on the 4 × 4 km grid of the NFI, covering the whole country area and all land uses, including afforestation, which considers national criteria for forest land—minimum area, height of trees (at maturity), crown cover, etc. [35]. Each sampling plot represents nearly 400 ha of country area. National sectoral emissions, including those of the LULUCF sector, also cover the 1990–2021 period and are obtained from national greenhouse gas inventory (as of 2023) [36] prepared according to the IPCC Good Practice Guidelines [36].

### 2.2. Projections of Greenhouse Gas Emissions and Removals under Different Scenarios

Projections of GHG emissions and removals are calculated using the same methodology as for the national GHG inventory under the United Nations Framework Convention on Climate Change requirements, applying IPCC Good Practice Guidelines [37]. GHG projections are estimated according to 4 different land-use scenarios for the period of 2021–2025 and the years 2030 and 2050, taking into account the LULUCF accounting rules provided in the LULUCF Regulation No EU 2018/841 [38] and its amendment No. EU 2023/839 [5] for 2021–2025. Land-use area changes are projected according to either historical changes or policy documents and established goals related to land-use change (Table 1). Analyzed scenarios include the following:

- The business-as-usual (BAU) scenario (scenario I) contains the assumption that the recently observed forest area will increase to reach 34.4% forest-area coverage in 2030 and 34.5% forest-area coverage in 2050 (3200 ha annually) according to the national forestry sector development plan for 2012–2020 [39];
- The forestry development scenario (scenario II) includes the assumption of a significant forest-area increase (from 34.1% in 2021 to 35.1% in 2030 and 35.3% in 2050)—8000 ha annually, including both human-induced afforestation and natural forest expansion, as indicated in Policies and Measures and Projections of Greenhouse Gas Emissions in Lithuania [30];
- The forestry development + additional measures scenario (scenario III) makes additions to scenario II, including preliminary measures for increasing GHG removals and decreasing GHG emissions from the LULUCF sector as indicated in the Integrated National Energy and Climate Plan [40] and Policies and Measures and Projections of Greenhouse Gas Emissions in Lithuania [30]. All additional measures under this scenario are dedicated to the cropland, wetland and grassland categories (Table 2);
- The forest area 40% + additional measures scenario (scenario IV) takes into consideration a more pronounced afforestation rate according to the project of the National Forest Agreement [41], aiming at 40% forested land to be achieved by 2050 and the same measures as in scenario III (Table 2). This nonbinding forest-land expansion

could be achieved with a 13,200 ha annual forest-land increase and represents more ambitious employment of the LULUCF sector for climate neutrality goals.

To project GHG emissions and removals for forest land, projections of growing stock volume, increment and mortality on forest land remaining forest land as prepared by State Forest Service and applied in the Policies and Measures and Projections of Greenhouse Gas Emissions in Lithuania [30] are used. The aforementioned projections by the State Forest Service consider the age-class distribution in Lithuanian forests in the future and, due to the relatively large share of old forest stands, foresees a nearly 3% decrease in the growing stand volume increment in 2050, as well as an 8% increase in forest harvests in 2030 and 10% in 2050 [30]. These ratios of change are applied for projecting GHG emissions/removals for forest land and for estimation of carbon-stock changes in harvested wood products. To project growing stock volume changes in afforested land, areas are multiplied by the annual growing stock volume change, according to the function applied in the National GHG Inventory (as of 2021) [42].

The same criteria for the areas that could be converted to forest land in all scenarios are used. They include the fertility rate of agricultural areas (only nonfertile or abandoned agricultural areas can be afforested), limitations regarding existing drainage systems in agriculture, etc. [43].

**Table 1.** Description of scenarios.

| Land-Use Category | BAU | Forestry Development | Forestry Development + Additional Measures | Forest Land 40% + Additional Measures |
|---|---|---|---|---|
| Forest land (remaining) | $2.25 \times 10^6$ ha in 2030; $2.30 \times 10^6$ ha in 2050 | $2.29 \times 10^6$ ha in 2030; $2.45 \times 10^6$ ha in 2050 | | $2.34 \times 10^6$ ha in 2030; $2.59 \times 10^6$ ha in 2050 |
| | Growing stock increment: $19.76 \times 10^6$ m$^3$ in 2030, $19.48 \times 10^6$ m$^3$ in 2050; Growing stock change: $4.95 \times 10^6$ m$^3$ in 2030, $5.05 \times 10^6$ m$^3$ in 2050; Felling: $11.38 \times 10^6$ m$^3$ in 2030, $11.54 \times 10^6$ m$^3$ in 2050 Policies and Measures and Projections of Greenhouse Gas emissions in Lithuania [30] | | | |
| Land converted to forest land | 3.2 kha annually from grassland to forest land | 4 kha annually from grassland to forest land; 4 kha annually from cropland to forest land | | 6 kha annually from cropland to forest land; 7.2 kha annually from grassland to forest land |
| Cropland (remaining) | 16.25 kha of perennial cropland (as of 2019); 2.29 kha annual increase in certified organic cropland (2010–2019 average); 4.29 kha annual increase in no-tillage cropland (2010–2019 average) | | Additional measures, covering perennial, certified organic and no-tillage cropland, as described in Table 2 | |
| Land converted to cropland | 33.74 kha annually from grassland to cropland (2010–2019 average) | | | |
| Grassland (remaining) | Organic drained soils comprise 6.2% of total grassland area [42] | | | |
| Land converted to grassland | 36.66 kha annually from cropland to grassland (2010–2019 average) | | Additional measures, covering cropland conversions to grassland, as described in Table 2 | |
| Wetlands (remaining) | 13.83 kha of peat extraction (as of 2019) | | | |
| Land converted to wetlands | No new conversions projected | | Additional measures, covering cropland conversion to wetlands, as described in Table 2. | |
| Settlements (land converted to settlements) | 0.4 kha annually from grassland to settlements | | | |

**Table 1.** *Cont.*

| Land-Use Category | BAU | Forestry Development | Forestry Development + Additional Measures | Forest Land 40% + Additional Measures |
|---|---|---|---|---|
| Other land (land converted to other land) | No new conversions projected | | | |
| Harvested wood products | 8% increase by 2030, 10% increase by 2050; same ratio as in 2019 among the categories of sawn wood, wood-based panels and paper products | | | |

Harvested wood products (HWPs) are projected by applying a first-order decay function, as specified in the IPCC Guidelines [44], meaning that all wood products (sawn wood, wood-based panels, paper and paperboard), once produced, enter the HWP pool as an input (whole amount of $CO_2$ sequestered) and then gradually decay each subsequent year. The half-period of decay is 35 years for sawn wood, 25 years for wood-based panels and 2 years for paper products, meaning that each subsequent year after production, 1/35 of (remaining) sawn wood's $CO_2$, 1/25 of (remaining) wood-based panels' $CO_2$ and ½ of paper products' $CO_2$ is released back to the atmosphere. HWP carbon stock change is a balance between $CO_2$ input (with new products) and output (from the decay of previous products). HWP input includes both domestically consumed and exported products (sawn wood, wood-based panels, paper and paperboard) produced from domestically harvested wood; exported roundwood is not included in the calculations. The carbon stock balance in HWP for all 4 scenarios is projected by applying the same ratio among harvested wood products as in 2019 and considering projections of harvested wood volume. According to FAO [45], harvested wood products in Lithuania in 2019 consisted of $1.27 \times 10^6$ m$^3$ of sawn wood (56%), $0.85 \times 10^6$ m$^3$ of wood-based panels (37%) and $0.16 \times 10^6$ m$^3$ of paper products (7%) produced from a total of $6.67 \times 10^6$ m$^3$ of roundwood.

According to the IPCC Guidelines [37], conversion from one land use to another is considered to be effective for 20 years; therefore, at a certain time, the result of measures shifts from, for example, afforested land to managed forest land. A 20-year transition period is applied for all changes in the land-use categories in the projections; therefore, the effect of measures applied to increase carbon stocks might decline if new conversions are not projected.

**Table 2.** Additional measures included in projections of GHG emissions under scenarios III and IV, according to the Policies and Measures and Projections of Greenhouse Gas Emissions in Lithuania [30].

| Description of Measure | Affected Land-Use Category | Annual Area | Period Affected |
|---|---|---|---|
| Promotion of no-tillage crop management | Cropland | Gradually increasing to 800,000 ha in 2040 | 2021–2050, with the same ratio of area increase applied to 2041–2050 |
| Restoration of wetlands on arable peatlands and protection of perennial grass cover | Cropland, wetlands | Gradually increasing to 20,000 ha in 2040 Cropland converted to wetlands | 2021–2050, with the same ratio of area increase applied to 2041–2050 |
| Promotion of perennial crops (shrubs and trees) | Cropland | Gradually increasing to 26,300 ha in 2040 | 2021–2050, the same ratio of area increase applied to 2041–2050 |
| Promotion of perennial grassland management on organic soils | Cropland, grassland | Gradually increasing to 40,000 ha in 2040 Cropland converted to grassland | 2021–2050, the same area of grasslands on organic soils (converted from cropland) as in 2040 applied to 2041–2050 |
| Promotion of green bedding in agricultural land, planting of landscape elements on agricultural land | Cropland, grassland | Gradually increasing to 178,000 ha in 2040 (10% of arable land) Cropland converted to grassland | 2021–2050, the same area of grasslands (converted from cropland) as in 2040 applied to 2041–2050 |

Projected GHG emissions and removals are compared to the 2030 and 2050 GHG reduction targets as set in EU regulation (EU 2023/839) and the Lithuanian Climate Change Management Agenda [4]. It should be acknowledged that economic growth and other factors, such as the influence of climate change, are not addressed in the analysis. Potentially increasing biomass consumption due to bioeconomy development is partly covered in the projections in the form of increased harvest volume in 2030 and 2050, projected by the State Forest Service [30].

## 3. Results

### 3.1. GHGs Emissions, Removals and Land-Use Changes in 1990–2021

Over the period under analysis, overall GHG emissions in Lithuania decreased significantly (Figure 1). This more than double decrease has been mainly the result of significantly dropped energy consumption due to transitional decline, reforms and market restructuring after the country regained independence in 1990 [46]. In 1995, emissions reached 43% of the 1990 level. However, afterwards, the trend of national emissions shows no significant reductions, and a rather stable level of GHG emissions should be acknowledged.

As in the beginning of the analyzed period (67%), energy-related GHGs continued to constitute the largest share, though a decreasing share, of total countrywide GHG emissions over the period (in 2021—61%). There are some reasons behind this. First, until the pandemic situation and the energy crisis, final energy consumption had trended slightly upwards since the last economic crisis. Second, while the share of renewable energy sources increased from 17.2% in 2004 to 26.8% in 2020 in Lithuania [47], GHG emissions related to the energy sector decreased only 4.4% in the same period, mostly due to significantly increasing emissions from transport. From 2004 to 2020, transport GHG emissions increased by 55.5%, transport being the largest source of emissions in the energy sector—54.1% [36]. According to Eurostat data, renewables account for 21.28% of electricity and 46.63% in heating and cooling, but in the transport sector, renewables amount to only 6.46% as of 2021.

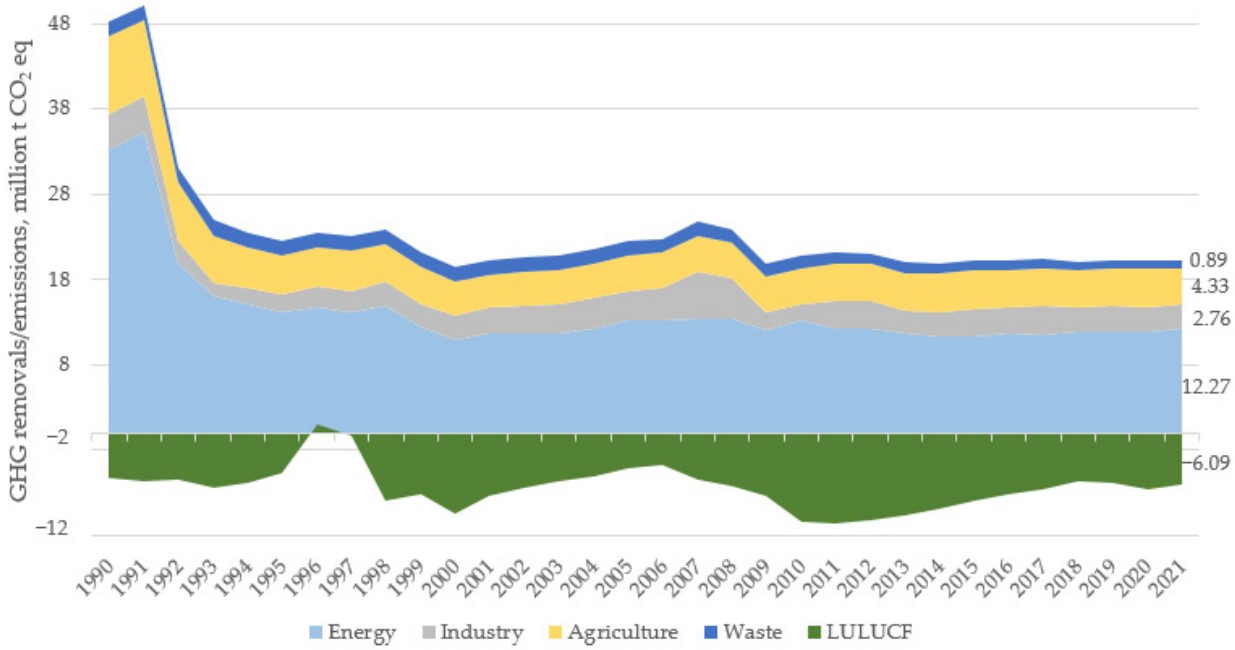

**Figure 1.** GHG emissions/removals in the LULUCF sector and total emissions in Lithuania during 1990–2021, million tons $CO_2$ eq (based on data from National GHG Inventory Report 2023).

The LULUCF sector has been a net sink of greenhouse gas emissions in Lithuania for almost the whole reporting period (1990–2021), except for 1996 and 1997, when, due to adverse natural conditions, LULUCF was recorded to be a net source of emissions (Figure 1).

Emissions in 1996 and 1997 are the result of repetitive droughts and consequent invasion by pests (e.g., *Ips Typographus*), which caused massive damage and death of spruce stands in Lithuania and therefore biomass losses from its forest land [42]. In addition to this, high emissions due to the drainage of organic soils (especially in cropland) also had an impact on the overall sector's net emissions. Emissions from drained organic soils varied from 1.9 in 1990 to 1.5 million tons of $CO_2$ eq in 2021 [36] due to conversions between land uses and different emissions factors (EFs) applied for different land uses. Though a sector's removal potential generally varies according to the natural conditions, economic factors are also of importance as they drive afforestation rates and use of agricultural land, as well as the volume of harvested wood products. Hence, an increasing area of grasslands converted from croplands increased GHG removals (starting after 2005); changing harvesting levels had an impact on both increasing and decreasing GHG removals, while an increasing growing stand volume increment, to some extent, compensates for the impact of increasing harvest levels. Though, in general, growing stand volume and harvest showed increasing trends over the analyzed period, the pattern of changes was different. Growing stock volume increased by 21.5% from 2007 to 2012, while afterwards, only 3.4% growth (2012–2020) was observed. The harvest level decreased by 16.8% from 2007 to 2012 and afterwards increased by 28.3% until 2020. Currently, 36% of wood is used for energy and 64% for materials (estimated from data in the National Greenhouse Gas Inventory [42] and FAO [45]).

The biggest change in the land-use categories was recorded for croplands (Figure 2), whose share shrank from 37% in 1990 to 31% in 2021. Correspondingly, forest area increased from 31% to 34% and grassland from 20% to 23%. Hence, land-use changes indicate some higher potential for GHG removal, as the area of cropland, which usually acts as a source, has decreased, initially due to the abandonment of cropland areas (which were gradually converted to grassland) after the restoration of independence and the subsequent economic recession.

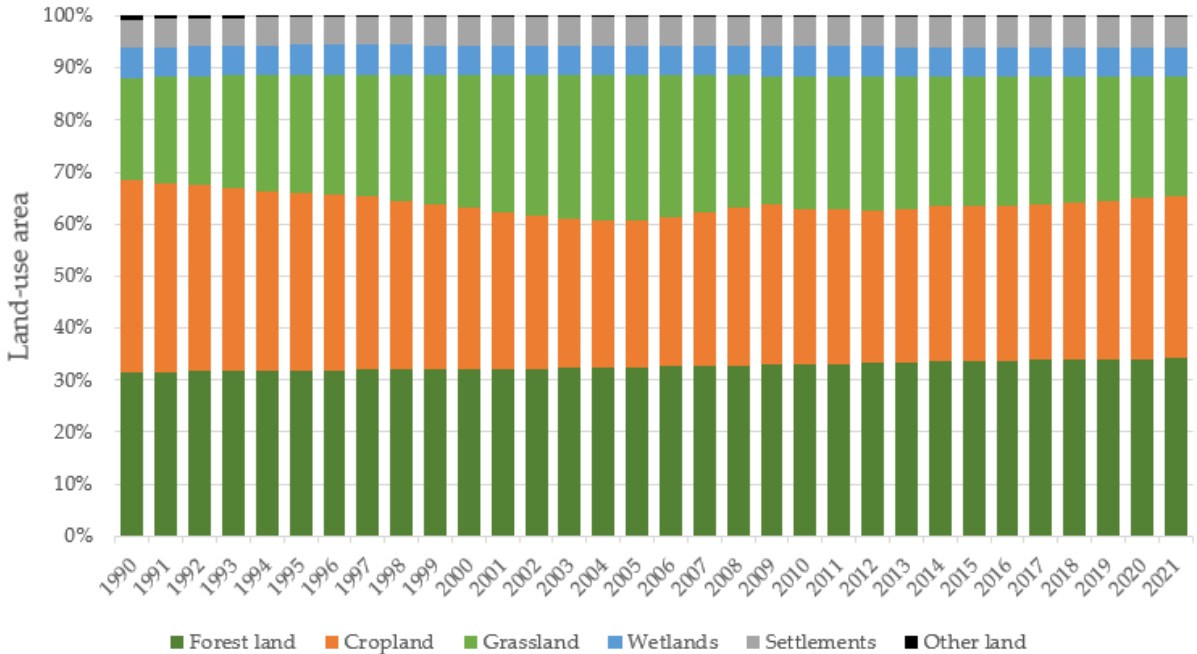

**Figure 2.** Land-use changes in Lithuania in 1990–2021 (based on data from National GHG Inventory Report 2023).

During the whole period of analysis, cropland acted as a source of GHGs, constituting 81% of the sector's GHG emissions at the beginning and 30% at the end of the period (Figure 3). The decrease in GHG emissions from cropland is related to the shift from traditional intensive agricultural practice to larger areas of no tillage crop practice,

organic agriculture, which is reported to increase soil organic carbon due to reduced soil disturbance and increased input of organic matter [36]. Decreasing overall emissions from cropland are partially outweighed by increasing emissions from wetlands, settlements and other land. The results also show the significance of the forest land category, as it provides the highest share of removals in the overall LULUCF balance (Figure 3). The forest category counterbalances emissions from cropland, wetlands and settlements and provides the potential to counterbalance other sectors' emissions altogether with harvested wood products (HWPs) and grassland. In 1990, forests accounted for 89% of the sector's removals; in 2021, 77%. The total amounts reached a maximum in 2011 with removal of some 10.173 million tons of $CO_2$ eq Though forests dominate removals, over the period of analysis, the share of grassland and harvested wood products in GHG removals also slightly increased.

If the years 1996 and 1997 are excluded, the land-use-related sector in Lithuania absorbed some 11–46% of the country's yearly emissions in 1990–2021 (Figure 1). This indicates that the LULUCF sector's foreseen offsetting potential (9.558 million tons of $CO_2$ eq), as needed for 2050, has been reached already in 2010, 2011 and 2012. Even bearing in mind the sector's uncertainties, this suggests some possibilities to reach the target set.

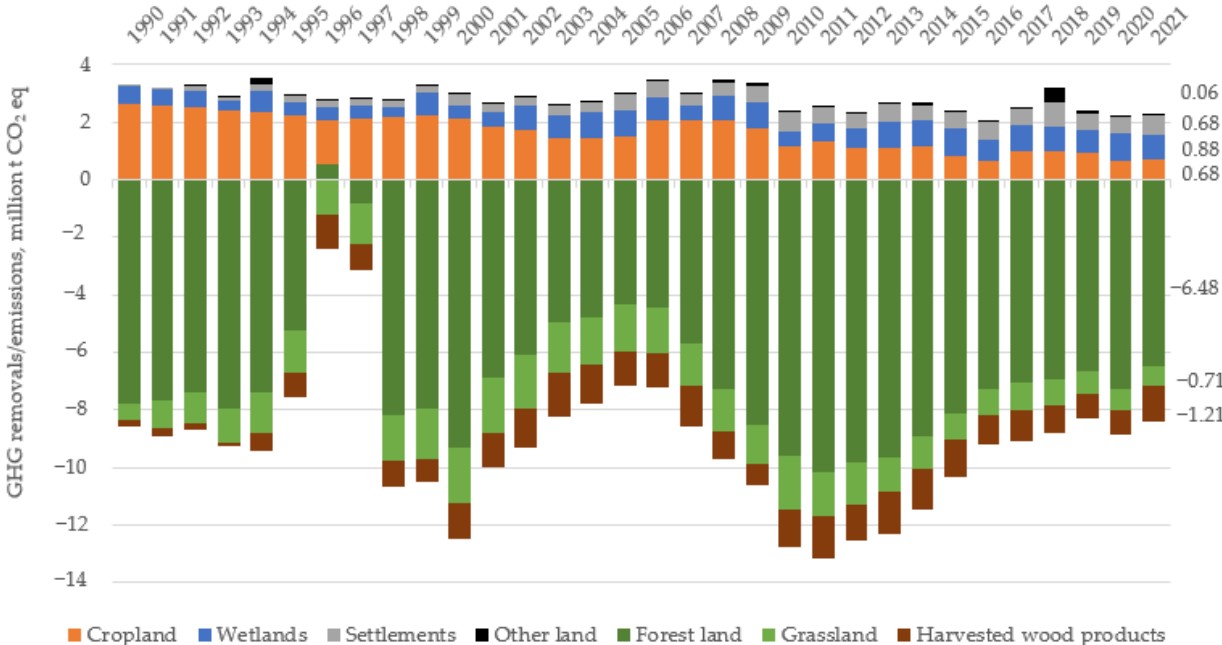

**Figure 3.** LULUCF GHG balance in 1990–2021 (million tons of $CO_2$ eq) (based on data from National GHG Inventory Report 2023).

### 3.2. GHG Projections for the 2021–2025 Period According to the Different Land-Use Scenarios

Though historical removals look favorable, after the EU LULUCF accounting rules (EU Regulation 2023/841 and its amendment 2023/839) [5,38] are applied, this aforementioned offsetting looks less promising. The maximum LULUCF credit allowance for Lithuania equals 6.5 million tons of $CO_2$ eq for 2021–2030 (EU Regulation 2023/857)) [48] (Table 3); however, only removals accounted for in 2021–2025 can be used for this purpose [5].

In the case of the BAU scenario (scenario I), 2.33 million tons of $CO_2$ eq in accounted-for removals by the LULUCF sector could be generated in 2021–2025, which is only 35.8% of the removals allotted to offset other sectors' GHG emissions. Accounted-for removals from LULUCF sector could amount to 2.41 and 4.53 million tons of $CO_2$ eq in the forest development (scenario II) and forest development + additional measures (Scenario III) scenarios during 2021–2025, correspondingly. It is evident that additional measures included in scenario III could have a significant impact on the accounted-for LULUCF GHG balance in 2021–2025. However, even in scenario III, the achieved accounted-for removals

are not sufficient to offset the desired amount of GHG emission from the other sectors. Furthermore, in this case, LULUCF could provide only some 70% of the amount allowed for offsetting, even considering that the amount allocated for offsetting (6.5 million tons of $CO_2$ eq) seems quite low—it is less than average annual sector's removals during the last 10 years of inventory. In addition, according to the unpublished data of the Ministry of the Environment of Lithuania, a shortage of approximately 6 million tons of $CO_2$ eq of annual emission allocations (AEAs) is expected in 2021–2030 if no additional measures for GHG emission reduction in non-ETS sectors are applied [49]. Hence, the results indicate that LULUCF might play a crucial role, despite being insufficient, for the implementation of climate change mitigation goals in the case of Lithuania in the short term to conform to the EU level commitments.

In addition, it is apparent that the LULUCF sector's contribution to climate change mitigation target achievement is significantly determined by the managed forest land category (Table 3), since the estimated managed forest reference level ($-5164$ kt $CO_2$ eq annually and $-25,820$ kt $CO_2$ eq for 5 years) for Lithuania [50] is higher than the projected forest land remaining forest land removals in 2021–2025. Therefore, application of the reference level leads to the accounted-for GHG emissions from this category in 2021–2025. Despite that, the afforested land category balances emissions with net removals in forest land (Table 3). The forest land 40% + measures scenario (scenario IV) shows that if the National Forest Agreement [41] were implemented, the LULUCF sector could provide additional GHG removals even in the short term (73.5% of the allowed offsetting). However, the greatest potential lies in the future. Afforestation plays a crucial role not only due to its high GHG removal potential but also because all GHG removals by afforestation can be accounted for as removals for offsetting (EU Regulation 2018/841 and its amendment 2023/839) [5,37] until the end of the conversion period of 20 years, when afforested areas are shifted to the managed forest land category. However, afforestation rates may be limited by the Nature Restoration Regulation [51] in areas where different land-use restoration may be required; thus, reported and accounted-for GHG removals from afforested land could be smaller.

**Table 3.** Reported and accounted-for 2021–2025 GHG emissions/removals (per EU Regulation 2023/839) [5] and LULUCF flexibility limit for 2021–2030 (per EU Regulation 2018/842 and its amendment 2023/857) [48,52], kt $CO_2$ eq ("IE": included elsewhere—in managed forest land reference level).

| Accounting Category | Reference Values (Annual) | BAU Scenario | Forestry Development Scenario | Forestry Development + Measures Scenario | Forest Land 40% + Measures Scenario |
|---|---|---|---|---|---|
| | | 2021–2025 | 2021–2025 | 2021–2025 | 2021–2025 |
| Managed forest land | −5164.64 | −24,716.13 | −24,530.24 | −24,530.24 | −24,530.24 |
| Afforested land | - | −5417.84 | −5668.39 | −5668.39 | −5906.76 |
| Deforested land | - | 590.67 | 590.67 | 590.67 | 590.67 |
| Managed cropland | 841.9653 | 4249.97 | 4237.89 | 2119.67 | 2119.67 |
| Managed grassland | −1210.13 | −837.05 | −833.12 | −843.28 | −865.84 |
| Managed wetlands | 791.9271 | 4083.10 | 4083.10 | 4087.50 | 4087.50 |
| Harvested wood products | IE | −3988.86 | −3988.86 | −3988.86 | −3988.86 |
| Balance (accounted-for GHG) | | −2331.73 | −2404.54 | −4528.52 | −4778.93 |
| Limit for offsetting | | −6500 | −6500 | −6500 | −6500 |

### 3.3. GHG Projections until 2030 and 2050 According to the Different Land-Use Scenarios

To assess the potential of LULUCF for climate mitigation in Lithuania until 2030 and 2050, projections based on the LULUCF reporting guidelines set forth in the IPCC Good Practice Guidelines [37] and the LULUCF accounting rules set forth by the EU in LULUCF Regulation [5,38] were carried out.

The results (Table 4) show that projections of GHG balance vary significantly and are sensitive to land-use changes, except for harvested wood products. Since forest stands in newly afforested areas will not reach maturity for harvest until 2050, projected forest land

expansion does not have an impact on carbon sequestration in harvested wood products. It is evident that newly afforested areas (due to afforestation and natural forest expansion) not only constitute a significant sink but also play a significant role in offsetting emissions from agricultural land uses. It could be stated that if Lithuania is able to maintain a stable land-use change pattern as observed in recent years (small areas of deforestation, large afforested/reforested areas (32,000 ha annually) and increased conversion from cropland to grassland), a total of 5.74 and 6.31 million tons of $CO_2$ eq could be sequestered in the LULUCF sector correspondingly in 2030 and 2050. This could ensure compliance with and overachievement of the EU target for 2030 but could only partly (66%) reach the national target for 2050. In 2050, those numbers could amount to 7.70 and 8.1 million tons of $CO_2$ eq correspondingly in the forest development and forest development + additional measures scenarios. The latter implies that the amount expected to be offset by the LULUCF sector in 2050 would be closer to the target only if the additional measures included in scenarios II and III were applied (81% and 85% respectively). Meanwhile, the 2030 LULUCF targets would be exceeded in all four scenarios (Table 4).

**Table 4.** Projected GHG emissions/removals (kt $CO_2$ eq) and targets for 2030 (per Regulation 2023/839 (EU)) [5] and 2050 (per Lithuania's National Climate Change Management Agenda [4]).

| Land-Use Category | BAU Scenario | | Forestry Development Scenario | | Forestry Development + Measures Scenario | | Forest Land 40% + Measures Scenario | |
|---|---|---|---|---|---|---|---|---|
| | 2030 | 2050 | 2030 | 2050 | 2030 | 2050 | 2030 | 2050 |
| Forest land | −6403.0 | −7531.8 | −6466.2 | −8907.9 | −6466.2 | −8907.9 | −6681.7 | −9865.5 |
| Cropland | 912.4 | 1351.7 | 904.3 | 1327.5 | 83.3 | 1045.8 | 80.5 | 1034.9 |
| Grassland | −623.9 | −583.4 | −621.7 | −575.8 | −628.9 | −749.2 | −636.3 | −754.4 |
| Wetlands | 816.6 | 816.6 | 816.6 | 816.6 | 836.7 | 878.7 | 836.7 | 878.7 |
| Settlements | 324.4 | 131.2 | 329.9 | 133.3 | 329.9 | 133.3 | 329.9 | 133.3 |
| Other land | 12.3 | 0.0 | 12.3 | 0.0 | 12.3 | 0.0 | 12.3 | 0.0 |
| Harvested wood products | −779.4 | −497.3 | −779.4 | −497.3 | −779.4 | −497.3 | −779.4 | −497.3 |
| Balance | −5740.7 | −6312.9 | −5804.2 | −7703.6 | −6612.3 | −8096.7 | −6838.0 | −9070.3 |
| Target | −4633 | −9558 | −4633 | −9558 | −4633 | −9558 | −4633 | −9558 |
| % of the target | 124 | 66 | 125 | 81 | 143 | 85 | 148 | 95 |

Although the targets for 2050 will be not reached, in all scenarios, higher total removals will be achieved in the long run (2050) except in the cropland category, where higher emission reduction will be achieved in the short term, and in the harvested wood product category, with declining removals in the long term (Table 4). The most significant effect of the measures applied can be observed in the cropland and grassland categories in 2030 (Table 4). Conversion of conventional agricultural land to no-tillage crops will be the most intense until 2030, therefore causing the most significant effect on carbon sequestration (in soils) until 2030. In addition, restoration of wetlands will result in a slight decrease in emissions for the cropland category at the expense of restored drained areas but will contribute to the wetland GHG source. Restoration of wetlands via $CH_4$ would additionally result in 25 kt $CO_2$ eq in 2030 and 62 kt $CO_2$ eq in 2050. This suggests that measures should be thoroughly accounted for and considered before implementation, also taking into account different time perspectives regarding LULUCF's climate change mitigation potential. It should also be considered that the growing stock increment in mature stands is shrinking, and the areas of mature stands will increase in upcoming decades. The declining sink in old forest stands and the remaining high emissions from drained forest organic soils will be counterbalanced by significant removals in young forests—areas recently shifted from land converted to forest land to forest land remaining forest land. Therefore, expected afforestation might be not enough, and additional measures (or an increase in their volumes) for increasing carbon sinks or reducing emissions in other land-use categories

(where available) are needed to rely fully on the sector's potential for 2050. This is clearly seen in the case of scenario IV (forest land 40% + measures), which indicates that, at least, much more pronounced afforestation levels are needed to reach higher removals in the LULUCF sector by 2050 to approach the climate neutrality target. If the National Forest Agreement were implemented [41], it could be expected that some 95% of the target for LULUCF GHG removals would be achieved (Table 4). Hence, even in the case of the most significant forest expansion, a shortage of approximately 0.5 million tons of $CO_2$ eq in GHG removals (Figure 4) in the LULUCF sector is expected compared to what would be necessary to reach the climate neutrality goal. In all cases, this shortage must be covered either by the sector itself taking additional measures or with more pronounced reductions of GHG emissions in the other sectors.

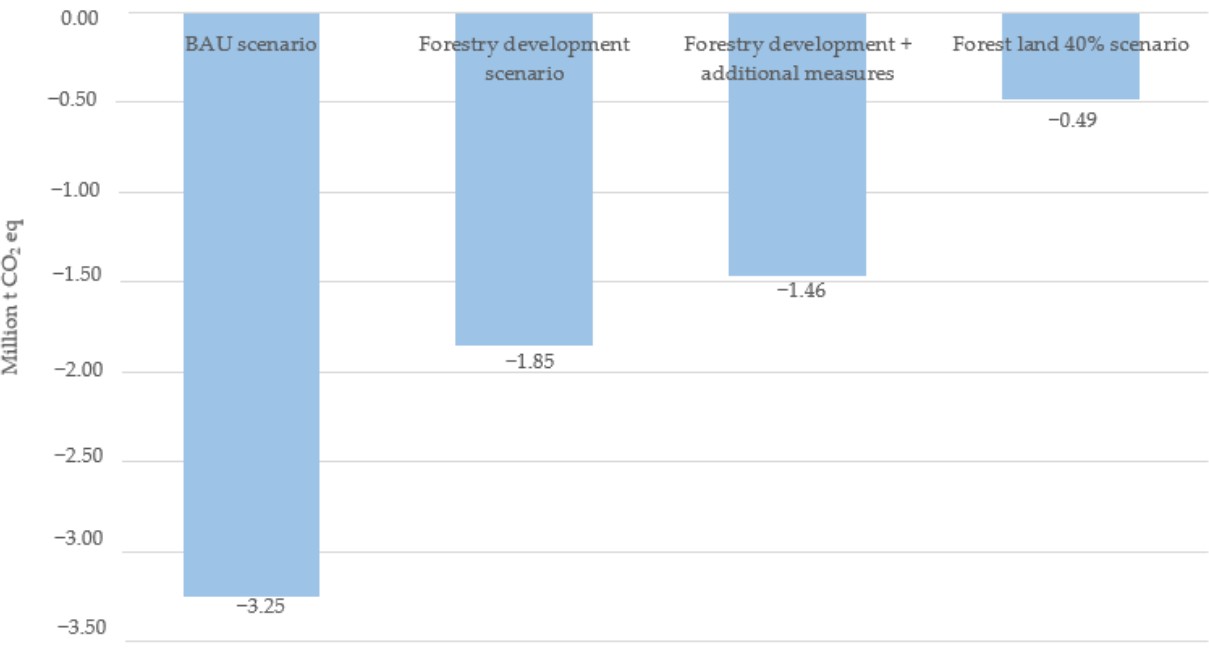

**Figure 4.** Shortage of GHG removals (million tons of $CO_2$ eq) in LULUCF in 2050.

### 4. Discussion

Since 1990, Lithuania has reduced its national GHG emissions more than 60%, but the challenge remains to meet the intended goals for 2050. Most of the decrease in GHGs resulted from a transformational decline in 1990–1994 after the fall of the Eastern Bloc, as energy consumption dropped more than twofold [46]. It should be admitted that not much more action has been taken than general EU regulations and market pressure have demanded. International and other commitments regarding GHGs have been achieved without significant efforts due to the very high emissions in the reference year. Hence, reducing GHGs in so-called non-ETS sectors remains the most problematic. The transport sector prevails as the most complicated one with the largest share in energy consumption and GHG emissions in Lithuania [36], with only 6.5% renewables (Eurostat data) and a relatively old car fleet in the EU [53].

In addition to the other economic sectors to be addressed to reduce GHGs, according to the National Climate Change Management Agenda [4], it is expected that LULUCF will remove at least 6.5 million tons of $CO_2$ eq over the period 2021–2030 and, in 2050, will offset 20% of the other sectors' remaining emissions compared to 1990. Historical trends provide some optimistic perspectives, as, over the total period of the analysis, LULUCF acted as a sink for 26% of national $CO_2$ eq emissions on average annually. However, though removals by the LULUCF sector are accounted for in all scenarios for 2021–2025 to offset emissions from the other sectors, application of the EU accounting rules results in much lower GHG removal potential for offsetting. According to the different scenarios, accounted-

for removals can reach only up to 36%, 37%, 70% and 74% of total amount allowed for offsetting by EU Regulation 2023/857 for 2021–2030. The role of accounting rules and the cap in achieving climate change goals also is acknowledged by other scholars [13,54], and the need for substantial changes in accounting rules is discussed [55]. Schlamadinger et al. [56] suggest that a fixed cap for forest-management accounting does not encourage countries to improve forest management unless a country is below the cap or faces such a risk. Considering that a significant share of Lithuanian forest stands are relatively old or will become old in the very near future, higher harvest rates are obvious; hence, the set forest reference level is not favorable in the case of Lithuania, as it encourages a reduction in harvest intensity (at least for 2021–2025, as set in EU Regulation 2018/841) [38] in order to preserve larger GHG removals in forest biomass. On one hand, a lower harvest rate would be preferable for forest-land carbon sink enhancement. On the other hand, it could be a solution for the short term only, since old forest stands have lower GHG removal potential due to their lower yield [57] or might even become a GHG source in the future. Therefore, the newest updates in the rules (EU Regulation 2023/839) [5], maintaining no specific accounting categories or reference values since 2026 and setting an overall GHG removal goal for the LULUCF sector for the first time, seems to be more beneficial for Lithuania. The results show that the 2030 mitigation targets could even be exceeded, and Lithuania could rely on LULUCF at least until 2030.

However, reaching the 2050 target remains more challenging. Lithuania would achieve only some 81% and 85% of the desirable 20% offsetting of 1990's GHG levels via LULUCF in 2050 in the forest development and forest development + measures scenarios. Hence, additional measures or changes in their volumes are still needed either to increase LULUCF potential or reduce emissions more significantly from the other economic sectors. The effect and the continuity of the measures proposed should be considered to yield a substantial number of credits in the future because the given measures might have different results in the short and long term, as our results indicate. Additionally, different environmental goals might intervene, such as wetlands restoration and climate change mitigation.

Forest land is the main reporting category for carbon removal in Lithuania. It could play a more pronounced role in climate mitigation if the National Forest Agreement [41] is implemented and forest land area reaches 40% of country area by 2050. Projections show a significant input of afforestation for climate change mitigation. The scenario including more extensive forest development (40% of total country area) indicates that, in this case, climate mitigation goals for 2050 could be achieved by 95% for LULUCF in Lithuania. Hence, potentially more ambitious goals for afforestation should be set instead of the current 3200 ha or 8000 ha planned annually. Other studies also report the significant influence of afforestation on carbon sequestration (for e.g., [11]). Nevertheless, taking into account the recently decreasing ratio of forest expansion (both natural forest expansion and afforestation) in Lithuania, the business-as-usual scenario may also be challenging to maintain. Natural forest expansion has been the most important source of the forest coverage increase in Lithuania, and according to the results of the State Audit [58], there are still large areas of natural forest expansion that are not included under forest land and are at risk of being clear-felled and used again for agricultural purposes. In addition, as historical trends indicate, natural disturbances or economic factors might also influence forest development and the potential to remove GHGs.

Climate change and bioeconomic inconsistencies might also play an important role in reliance on the LULUCF sector's climate mitigation potential. Trade-offs between bioeconomy (forest biomass harvesting) and carbon sequestration [59] should be considered in the light of climate mitigation goals, as an increase in biomass removals might lead to forests becoming a carbon source rather than a sink in the future [60]. Nevertheless, Kauppi et al. [61] suggest that timber harvesting and carbon sequestration can be aligned if proper management of forests is applied. Though the influence of bioeconomic development is not analyzed in detail in this paper, results show that a projected 10% harvest removal increase

will have no negative impact on GHG removals as growing stand volume is still increasing in the case of Lithuania.

Some other uncertainties within the sector also should be considered [62], and saturation of sequestration capacity and the vulnerability of the sector should be taken into account when relying solely on forests and LULUCF in general [13]. For example, the forest management cycle [63] and other forest and agroecosystem management decisions [64] may have an impact on carbon sequestration and, thus, climate change mitigation potential in the LULUCF sector as well. Forests' climate change mitigation potential might also be affected negatively by climate change (see, for example, [65]), and forests might become carbon sources instead of sinks [66,67].

## 5. Conclusions

Despite this sector's vulnerability to natural disturbances and uncertainties, the LULUCF sector in Lithuania shows significant potential for carbon sequestration, with most of the removals occurring in forest land, followed by harvested wood products and sequestration in grassland. GHG removals by forest land not only ensure this large sector's removal potential but also ensure its ability to counterbalance other sectors' emissions. Though the "no debit" rule set in LULUCF regulations is to be met and it will even be possible to generate accounted-for removals in the LULUCF sector during 2021–2025 as well as the 2030 target for LULUCF, reaching carbon neutrality in 2050 will be challenging for Lithuania. Along with afforestation, the analyzed measures for increasing LULUCF potential are insufficient and must be considered while taking into account the benefits in the short and long term—additional measures in line with planned forestry development could achieve only some 85% of the needed GHG removals for 2050. In addition, even rather ambitious afforestation goals (an increase in forest land to 40% of total country area) are not enough and, with other measures applied, could ensure only 95% of the national climate neutrality target for LULUCF. Moreover, afforestation rates could be limited by the lack of areas suitable for afforestation due to national criteria. Additionally, the requirements set by Nature Restoration Regulation (2022) might influence land-use change patterns and the GHG balance of the LULUCF sector or separate land-use categories. Hence, reconsideration of afforestation targets and of other measures is needed if LULUCF is to be relied on for climate neutrality implementation. In addition, the sector's peculiarities and uncertainties are of importance and must be considered by policy makers while pursuing national climate neutrality and climate change mitigation in general.

**Author Contributions:** Conceptualization, R.D. and V.K.; methodology, V.K., R.D.; software, V.K.; validation, R.D., V.K.; formal analysis, R.D.; investigation, V.K.; resources, V.K.; data curation, V.K., R.D; writ-ing—original draft preparation, V.K.; writing—review and editing, R.D.; visualization, R.D.; su-pervision, R.D. All authors have read and agreed to the published version of the manuscript.

**Funding:** This research received no external funding.

**Data Availability Statement:** Data used in the research will be made available upon request.

**Conflicts of Interest:** The authors declare that they have no known competing financial interests or personal relationships that could have appeared to influence the work reported in this paper.

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
