# Peer review of "Impact of Land-Use Changes on Climate Change Mitigation Goals: The Case of Lithuania"

_land, doi:10.3390/land13020131_

Round 1

Reviewer 1 Report

Comments and Suggestions for Authors

The article is discussing important subject of reliability of the neutrality targets in LULUCF, pointing out complexity of the task and necessity to implement significant changes in the land use strategy to fulfill the proposed targets, while the article also clearly demonstrates that there is possibility to reach this target. However, the article can also be improved. One of the shortcomings is missing information about the largest sources of emissions in LULUCF sector. I can imagine that the largest source is organic soils, which should be mentioned in the article.

In methodology, chapter 2.1 it is noted that Land use change is important parameter; however, accounting of it in Lithuania differs from the most of the countries, so this property can be explained more in details, particularly, accounting of afforested lands.

In chapter 2.2 description of afforestation should expanded, which areas will be afforested in each scenario, what are criteria for afforestation - lands to be converted. In the same chapter, lines 149-160, I think it is better to use intensity parameter, showing, how many wood is utilized from the resources available for felling. In lines 162-181 or somewhere else in methodology it should be noted, how exported roundwood is considered in calculation, particularly, in afforested lands to ensure that the projections are in line with the GHG inventory production approach. In lines 176-181 and other places referring to the transition period - I m not sure if this has any practical value, since afforestation and deforestation are accounted separately only till 2025.

Table 2 would benefit from additional information on what is source of land for land use changes, respectively, initial land use. In lines 185-191 or other place it should be noted that increase of harvest rate basically means significant increase of forest biofuel supplies.

Before or after Figure 2 it might be valuable to explain the reasons for reduction of emissions from cropland, since it looks surprising considering increase of farm production.

In relation to Table 3 it should be noted in the article that the scenario analysis do not consider potential negative (or positive?) effect of nature restoration regulation prohibiting management of certain forest area and limiting area available for afforestation.

In chapter 3.3. it should be explained if this assumption is not contradicting to projections of agricultural production. I guess in real life things goes opposite and grassland are converted to cropland to fulfill requirements of integrated farming and organic farming.

In Table 4 it should be explained how the constant rate of HWP can be explained considering increase of forest area? Isn't it a mistake.

Reviewer 2 Report

Comments and Suggestions for Authors

Manuscript ID: land-2809725

Title: Impact of Land Use Changes for Climate Change Mitigation Goals: Case of Lithuania

Abstract:  The authors started the abstract with abbreviation without definition (All abbreviation should be defined when used for the first time)

The methodology used for LULUCF analysis is not introduced in the abstract.

Introduction: Information provided from line 32 to 40 requires citations.

Results are fine and to the standard.

Discussion is sound

Conclusion can be improved. Conclusion can be written without repeating results.
